# Relationship between Children’s Lifestyle and Fear during Dental Visits: A Cross-Sectional Study

**DOI:** 10.3390/children10010026

**Published:** 2022-12-23

**Authors:** Carolina Caleza-Jiménez, Mª del Mar López-de Francisco, Asunción Mendoza-Mendoza, David Ribas-Pérez

**Affiliations:** Department of Pediatric Dentistry, Faculty of Dentistry, University of Seville, 41080 Seville, Spain

**Keywords:** child, behavior, dental fear

## Abstract

The purpose was to analyze the relationship between new family and social patterns and child emotions in the dental clinic. The sample consisted of 174 children between the ages of four and nine. Parents completed 20 questions that referred to social, family, cultural, and economic factors. The Wong–Baker scale was completed by the children after the end of the treatment. A total of 129 of the children were found to be extremely happy (45.4%) or happy (28.7%) with the dental care received. There were statistically significant differences (*p* < 0.05) between younger children and sad and incredibly sad faces on the Wong–Baker test with a strong association (V > 0.3). Regarding children who practiced team sports, the chi-square test revealed a very significant association with happy and incredibly happy faces (*p* < 0.001) and the Cramer statistic evidenced a strong relationship between team sports and less dental fear (V > 0.3). Considering the limitations, small children (4–6 years) would be more fearful. In addition, stable environments in the family life of children (appropriate routines, adequate time spent with videogames, sport activities) could be interesting factors for improved control of fear and emotions in children. Further research is needed in this field.

## 1. Introduction

Dental fear and anxiety are common conditions that affect oral health and clinical management in individuals of all ages, although they appear to develop primarily in childhood and adolescence [1]. Fearful children are typically less cooperative and have poorer oral health since they try to avoid going to the dentist [2]. The terms ‘dental fear’ and ‘dental anxiety’ are almost identical, although a slight difference in meaning is found in the scientific literature [3]. In effect, fear is a physiological, behavioral, and emotional response to a frightening situation, while anxiety refers to distress and concern about having to face a frightening situation. However, the two terms are often used interchangeably for strong negative emotions associated with dental treatment [4].

A better understanding of the factors underlying dental fear can help pediatric dentists plan adequate treatment of patient behavior and offer better treatment strategies [5].

The reactions of children to the environment of dental treatment are diverse and complex [6]. Dental fear at an early age is a dynamic phenomenon in which many contributing factors intervene [5,7,8]. There is almost complete agreement in the literature that smaller children are more fearful than older children [3,9]. Other crucial factors to consider are previous negative experiences with dental treatment, and in this sense, the first experience a child has with dental treatment is especially important [3,7,10,11].

However, fewer studies can be found on the relationship between dental fear and family and social factors in children. In this sense, there have been changes in society and the routine habits of children that can influence on their behavior in the dental clinic [12].

There are increasing numbers of mothers who remain in employment. It often causes parents to have to leave their children in the hands of caregivers. This can result in excessive permissiveness, leading children to refuse to follow rules and to reject any kind of authority, including that of the dentist [13]. Other children adopt learned defenselessness behaviors, acting as victims when they have to visit the dentist to gain greater attention and induce feelings of guilt in their parents [14].

There have also been other social changes, with the appearance of new family models and a great increase in separations and divorce, and these factors can also influence childhood behavior [15]. In these situations, some children develop emotional disorders that their parents are unaware of. This and the fear of having to go to the dentist implies that some children are extremely difficult to treat [14].

Regarding the socioeconomic and educational level of the family, some studies have reported greater dental fear among children belonging to lower income families [5,16,17,18].

Lastly, it should be noted that technological developments have caused children to become more sedentary, as their form of entertainment now usually centers on videogames and watching television [19,20]. This is of interest, as some studies have associated fear with spending a lot of time in front of a screen [21].

Due to the new family and social patterns and considering the many children who require behavioral management in pediatric dental practice, our study aimed to evaluate and compare the influence of family structure, socioeconomic level, routine habits, and child social relationships on emotions in dental clinic.

## 2. Materials and Methods

The present study was carried out using a questionnaire designed and validated by experts in the field to study family structure, habits, social relations, and cultural/economic status.

The initial survey with 11 questions was previously conducted by four dentists (associated professors of pediatric dentistry at the University of Seville) who modified the tool proposed by Alarcon et al. [22] (Table 1). It received elevated levels in all questions for coherence, clarity, and relevance, but there were some suggestions for more questions.

Once validated, the pilot phase was carried out by conducting a survey of 15 parents. There were some questions that generated doubts when answering them, and we made some modifications. Most parents responded to the questions about their employment status, but did not specify the profession. Therefore, they were separated into two different questions. The question about marital status was also modified because there was no option aimed at those parents who share a family home without being married.

The final questionnaire consisted of 20 questions referring to four dimensions of the study (Table 2). Age categories were divided into two groups for preschool children (4–6 years) and school children (7–9 years) according previous studies [23,24]. Socioeconomic status, based on four questions (professional activity and educational level of fathers and mothers), was assessed using the four factors Hollingshead index. This status is based on the formula ((Occupation score × 7) + (Education score × 4)) [25]. To simplify the different populations of children included in this study (big cities, towns, villages, etc.) were split into +/− 30,000 inhabitants (based on the means of the different populations).

Parents in the waiting room completed the questionnaires while the children were attended by M.L., except for the self-reported adapted version of the Wong–Baker faces scale [26], which the children completed (Figure 1) to observe their emotions. Based on the study by Mendoza et al. [27], because pain perception has an important affective component (depression, fear, anger, anxiety, etc.), each child had to choose the face of this scale that best illustrated his experience that day according to the fear he would have had in the dental clinic. It consists of five faces with different facial expressions ranging from laughing to crying. The faces were explained to represent an extremely happy person (face number 1), a happy person (face number 2), a serious person (face number 3), a sad person (face number 4), and an extremely sad person (face number 5), depending on the fear they had felt in the dentist. This scale was used after the end of the previously planned conservative treatment (dental filling/pulp treatment) and before offering the children any reward for their behavior during treatment, to avoid influencing their response.

The sample size was calculated based on a unilateral hypothesis in contrast with a statistical power of 80% and a 95% confidence interval, as estimated in previous pilot studies. A 10% error margin was considered to compensate for lost subjects or missing data. The resulting sample size was 174 children aged 4 to 9 years, who attended the University Dental Clinic, most of them from the provinces of Seville and Cádiz (Spain). Patients who met the inclusion criteria were assigned a study number to collect data (Table 3). This age range was decided so that they would have enough ability to follow instructions and would not have adequate cognitive development from the child to adolescent period, where successful coping mechanisms were developed to possible previously stressful dental stimuli.

The responses were entered into an MS Excel spreadsheet to generate the study database, which was checked for errors or incomplete responses. A total of eight questionnaires were discarded because they did not complete more than half of the items on the questionnaire and marked multiple options.

The SPSS version 25 statistical package was used to perform the descriptive analysis, calculating the frequencies for each variable. Contingency tables (2 × 2 and 3 × 2) were generated to evaluate statistical associations between variables, based on the chi-square test. Statistical significance was set for *p* < 0.05. Furthermore, the Cramer statistic was used to assess the strength of the statistical association between the variables—an intense association defined by values of V > 0.3 and very intense for V > 0.6.

## 3. Results

There were no missing data for any of the study variables. Briefly, 129 of the children chose an incredibly happy face (45.4%) or a happy face (28.7%) after the dental treatment was received (Figure 2).

We have reported in Table 4 the associations between the influence of the family structure, socioeconomic level, routine habits, social relationships, and the Wong–Baker faces scale.

### 3.1. Influence of the Family Structure of the Child

There were no statistically significant gender differences in terms of sad or extremely sad responses. Regarding age, smaller children aged 4–6 years were found to be significantly sadder when they had to go to the dentist (*p* < 0.05) (Figure 3), and in the Cramer statistic was found an intense association (V > 0.3). Although 47.8% of the patients who chose sad or extremely sad faces were younger children in the families, no significant association was recorded. Similarly, no statistically significant associations were observed with respect to the different marital states.

### 3.2. Influence of Socioeconomic Level

Patients who belong to families of medium (34.7%) or low socioeconomic level (65.2%) produced many more sad or incredibly sad responses than in the group of high socioeconomic level (0%), although statistical significance was not reached. Likewise, no significant association was found with the population size of the city of residence or the type of school attended.

### 3.3. Influence of Routine Habits of the Child

A statistically significant association was recorded between patient bedtime and the Wong–Baker scale, with a better response to dental treatment among children who went to bed between 8 and 9 p.m. than among those who went to bed later (*p* < 0.05). Similarly, a significant association was observed between video games and a poorer attitude toward the dentist, with greater fear among children who played such games than among those who did not, although no correlation was observed with the time spent playing video games (Figure 4). Although children who watched cartoons (television or tablet) for more than half an hour a day produced sad (72.2%) and extremely sad faces (80%), no statistically significant associations were observed.

### 3.4. Influence of Social Relationships

No statistically significant associations were recorded in terms of whether the children participated in after-school activities or not. A total of 75.2% of the children played sports and, of these, 71.7% participated in team sports. The chi-square test revealed a highly significant association between these two variables and the Wong–Baker scale (*p* < 0.01 and *p* < 0.001, respectively). Furthermore, the Cramer statistic showed an intense relationship between team sports and less dental fear (V > 0.3) (Figure 5). No significant associations were recorded in terms of whether the children had a caregiver or not. Similarly, no significant associations were recorded in terms of whether the children had a domestic pet.

## 4. Discussion

Patient fear is the most widespread problem found in pediatric dentistry and is a challenge for the healthcare professional. Invasive treatments were among the main direct factors more likely to cause dental fear. When confronted with a stressful dental stimulus, a child could develop a reflexive psychological defensive reaction to it in the form of fear appearance [1,28]. The reason for this appearance is the fact that these direct invasive stimuli threaten to disrupt the integrity of the organism [29].

Another classic mechanism of the appearance of dental fear is the non-visiting of dental offices, a global problem of modern dentistry in the context of mechanisms for avoiding stressors within them. The formation and development of a vicious circle, associated with past medical experiences or assumed stressful situations within the dental office, leads to the development of dental fear, behavioral problems, and avoidance of dental visits from an early age [30,31,32,33]. This is the core problem of already impaired oral health in childhood, where parents have to assume more responsibility and play more important roles, not only in managing better oral health in their children (proper oral hygiene, anticariogenic diet, fluoride use) but also in cooperation for the conduct of better cooperative behaviors of their children in the dental office. Thus, the avoidance of visiting the dental office would decrease significantly over time [29].

Furthermore, society has experienced changes, with the appearance of new family models in which there are children who suffer emotional problems, which in turn are reflected in the visit to the dentist. The present study was carried out with the main aim of examining new factors in the child’s environment that can help us understand the emotions of our patients.

In the literature, the age and gender of the patient have been the most frequently evaluated factors, and younger age is associated with increased fear of the dental clinic [34,35,36,37]. This is consistent with the fact that the cognitive capacity of the child increases with age, leading to better awareness and understanding [38]. Small children suffer fear due to unawareness and a feeling of abandonment [5]. In our study, children between 4 and 6 years of age presented significantly poorer responses than those between 7 and 9 years of age, in agreement with the recent systematic review published in 2020 by Murad et al. [5]. Fears are evolutive and change as the child grows and matures, the natural tendency is for them to gradually subside over time. Therefore, the results obtained in the present study coincide with the natural characteristics of childhood. There was a big difference between age subgroups (only 61 children of 174 at 4–6 years old), so there is no proper statistical interpretation. However, in terms of patient gender, different studies have reported that fear is greater among female patients [35,37,39], although in our series no significant correlation was found between patient gender and fear of the dentist.

In the first years of life, the family has a strong influence on the cognitive, personal, emotional, and social-affective development of the child [40]. Society has experienced changes and the family structure has become more diversified as a result [41]. There is no evidence that a family structure different from conventional standards has a negative impact upon children. However, when parents separate or divorce, and there is a poor relationship between them before, during, or after separating, children suffer and may develop psychological problems [42]. In our study, we did not observe a relationship between the marital status of the parents and the mood state of the child—although this may be because only 9.2% of the study sample consisted of children with parents who were divorced, single, widowed, or others. It would be interesting to conduct further research in this field with a larger sample of subjects belonging to families of this kind.

Regarding cultural factors and socioeconomic status, some studies [16,17,18] have found these parameters to influence whether a dental visit will take place earlier (preventive) or later. This may condition pain perception on the part of children and their learning patterns, which in turn influences the development of dental fear and its persistence over time. However, few studies on this issue are available and more research is needed to determine how family and cultural factors influence dental phobias [43]. In our study, and in coincidence with the results published by Felemban et al. [44], we did not observe significant differences in the degree of dental fear in children according to the profession or educational level of their parents. Secondly, it is also well known that parents are role models for their children. Their dental behaviors and attitudes (about oral health maintenance, attendance to the dental office, etc.) play a major role in the transfer of habits from generation to generation. If parents develop the presence of dental fear over time, there is great certainty that their children could consequently behave in the same way [45,46]. It is an interesting factor for future research.

It is interesting to mention that the American Academy of Pediatrics (AAP) endorses the guidelines of the American Academy of Sleep Medicine (AASM) and encourages parents to establish a sleep routine that allows their children to get the necessary hours of sleep at an early age [47]. In our series, we observed a statistically significant association between bedtime and dental fear; however, the age factor must also be considered here, since younger (pre-school) children need more sleep and, therefore, go to bed earlier. Chaput et al. [48], in a review of 62 studies, found that schoolchildren who slept longer were better at controlling their emotions. Based on this, it would be logical to assume that these children would be better at controlling their anxiety when visiting the dentist, since there is evidence that children who do not sleep well can suffer health problems (obesity, arterial hypertension, depression, headache, etc.) and behavioral disorders (irritability, concentration problems, etc.).

The APP also recommends avoiding television, computers, telephones, or tablets in the child´s bedroom, particularly at night. In addition, it recommends that screen time be stopped at least one hour before going to bed [49]. Despite this, many children have no rules regarding electronics use and can develop behavioral problems as a result. Although we did not observe a statistically significant relationship between exposure time and the results of the Wong–Baker face scale, we found that children who watched cartoons for more than half an hour had selected more sad and incredibly sad faces. This is consistent with the study published by Mobarek et al. [49], where longer exposure times to the electronic screen were associated with increased fear among children in the dental clinic.

Videogames can alter the behavior of the players, their emotions, and learning. There are contradictory findings in the literature on the possible association between aggressive video games and aggressive behavior on the part of players [50]. The AAP recommends setting limits, avoiding spending too much time with these games. Furthermore, it recommends that more educational video games are preferred over violent games, considering the age of the child [51]. In our study, a significant association was observed between fear in the dental clinic and video games, although the time spent with such games was not found to exert a significant influence. It would be interesting to conduct further studies on the use of videogames, differentiating those of a violent nature and assessing their influence in terms of increased fear of the dentist, to better understand the attitudes of children in the dental setting.

In turn, it is important to mention the influence of social relationships in the context of the momentous changes in lifestyle experienced in recent years. Current socioeconomic conditions (lowered birth rate, individualism, urban development, increased use of videogames, tablets, and smart phones, etc.) determine the environment in which children are going to socialize [52]. After-school activities are a part of the education of our children and are an immense help in their development and socialization. In this regard, special mention should be made of sports activities, as they offer excellent health benefits to children. A current major problem in childhood is obesity and, in this sense, sports activities contribute to maintaining ideal body weight [53]. In our study, we observed a significant influence of sports activity on patient attitude in the dental clinic. A statistically significant association was observed between participation in team sports and reporting of happy or extremely happy faces with the Wong–Baker scale. Dimech and Seiler [54] described similar findings. Children who participate in team sports learn to work with their companions as a group and, therefore interact with individuals with personalities different from their own, promoting mutual respect and cooperation. These children must learn to listen to both the coach and the rest of the team and to accept and respect those companions who are less skilled in the sport. Several studies have found that children who participate in team sports have less depression, better academic performance, greater self-esteem, and fewer health problems in general [55,56].

This study has limitations, although we have investigated a wide range of aspects that can influence the attitudes of pediatric patients toward the dental visit, based on an adequate sample size. Many of the variables analyzed have not been studied in the dental field and, therefore, it has not been possible to establish comparisons with other studies. Furthermore, some of the results obtained were weak since they referred to only a small number of patients within the global sample. In addition, the types of treatments and the number of treatments can introduce bias in the results.

Ultimately, it is important that there were no statistical conditions and no multivariate analysis was undertaken, for this reason it is highly likely that bias is introduced because of confounding. It would be interesting for future studies to get an appropriate analysis with a better interpretation and formulation of conclusions.

Lastly, we recommend further research involving larger sample sizes to address these family and sociodemographic aspects of children. Society is changing and new family patterns, together with current lifestyle habits, can help us to understand the fears of children in our practice and know how to help them afford the best dental care possible. Efforts should focus on consolidating a stable environment in the family life of children, with adequate routines and healthy habits that contribute to improving their emotional development.

## 5. Conclusions

Taking into account the limitations of the present study, smaller children (4–6 years) would be more fearful, and in this study, significantly sadder when they had to visit the dentist.

Sleep routine and good nighttime rest are interesting factors for better control of fear and emotions in children—our results demonstrate a statistically significant association between earlier bedtime for patients and less fear of dental treatment.

Technological developments have made videogames extremely popular among children, and this phenomenon could be significantly associated with fear regarding dental treatment.

Sports activity and being part of a team promote respectful interactions with others, with fewer behavioral problems and greater self-esteem. This in turn would offer significant benefits in terms of patient attitude towards dental visits.

More research with multivariate studies is necessary to reach stable conclusions in this field.

## Figures and Tables

**Figure 1 children-10-00026-f001:**
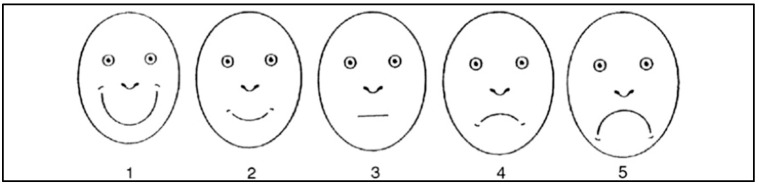
The Wong–Baker face scale.

**Figure 2 children-10-00026-f002:**
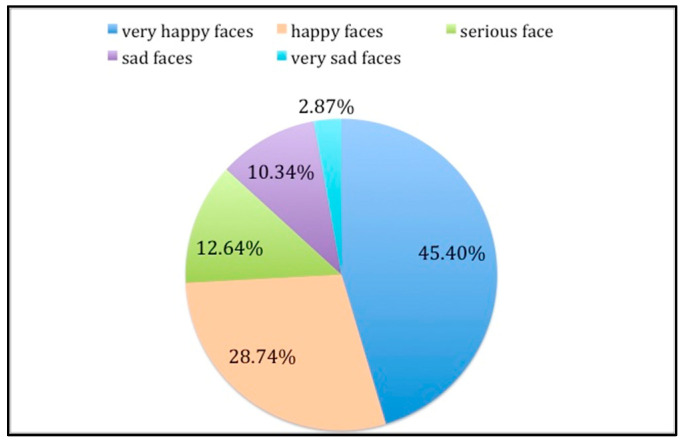
Pie chart of the faces chosen with the Wong–Baker faces scale.

**Figure 3 children-10-00026-f003:**
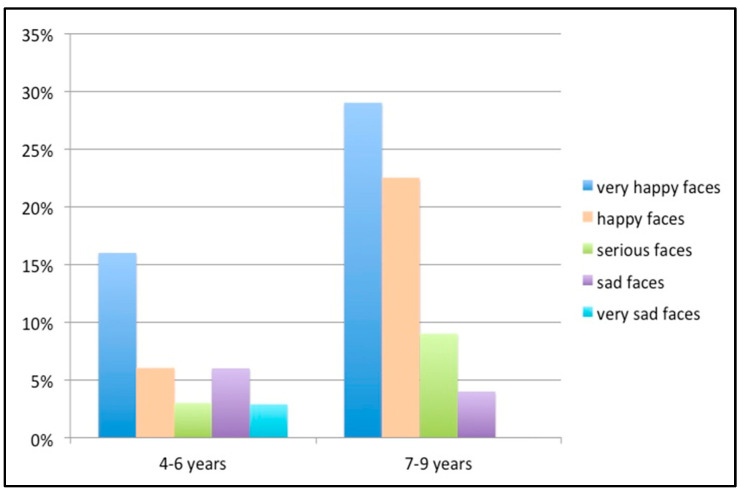
Bar chart of the results referring to the age of the patient.

**Figure 4 children-10-00026-f004:**
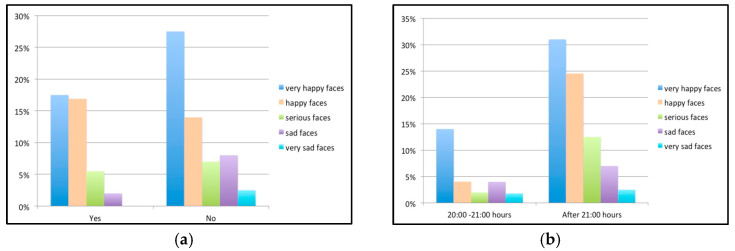
(**a**) Bar chart of the results referring to videogames; (**b**) results referring to bedtime.

**Figure 5 children-10-00026-f005:**
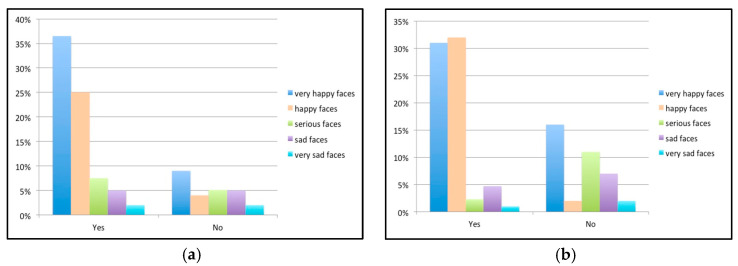
(**a**) Bar chart of the results referring to the practice of sports; (**b**) results referring to the practicing of team sports.

**Table 1 children-10-00026-t001:** Initial survey for validation.

INDICATORS	It doesn’t meet the criteria	Low level	Moderate level	High level	It doesn’t meet the criteria	Low level	Moderate level	High level	It doesn’t meet the criteria	Low level	Moderate level	High level
	COHERENCE	CLARITY	RELEVANCE
1.What is the sex of the child?-man-woman												
2.What is the age of the child?-4 years old-5 years old-6 years old-7 years old-8 years old												
3. Is a biological son?-Yes-No												
4. Is a child alone?-only son-eldest son-youngest son-other												
5. What is the occupation of the mother?-senior executive, professional, or owner of a large company-administrative or owner of a small company-shop assistant or technician-skilled worker-semi-skilled worker-worker without specialization												
6. What is the occupation of the father?-senior executive, professional, or owner of a large company-administrative or owner of a small company-shop assistant or technician-skilled worker-semi-skilled worker-worker without specialization												
7. What is the marital status of the parents?-married-divorced/separated-single-widower-other												
8.What studies has the mother?-university graduate-bachelor-secondary education-primary education												
9. What studies has the father?-university graduate-bachelor-secondary education-primary education												
10.How many inhabitants have the population of family residence?-0–5.000-5.000–30.000-30.000–100.000-more than 100.000												
11.What is type of school?-private-public-concerted												
SUGGESTIONS	

**Table 2 children-10-00026-t002:** Frequencies of the variables of the questionnaire administered to the parents.

Dimensions Studied	Sample (n = 174)	Dimensions Studied	Sample (n = 174)
Family structure		Habits	
Age		Bedtime	
4–6 years	61 (35.1%)	8–9 p.m.	41 (23.5%)
7–9 years	113 (64.9%)	>9 p.m.	133 (76.4%)
Gender		Videogames	
Male	87 (50%)	Yes	70 (40.2%)
Female	87 (50%)	No	104 (59.7%
Biological offspring		Time spent with videogames	
Yes	173 (99.4%)	0–30 min	23 (13.2%)
No	1 (0.6%)	>30 min	74 (42.5%)
Order among siblings		Time spent watching cartoons	
Only child	35 (20.1%)	0–30 min	53 (30.4%)
Older	48 (27.6%)	>30 min	121 (69.5%)
Younger	75 (43.1%)	Social relations	
Other	16 (9.2%)	After-school activity	
Parent marital status		Yes	99 (56.8%)
Couple	158 (90.8%)	No	75 (43.1%)
Divorced, single, widowed, other	16 (9.2%)	Sports	
Cultural and economic status		Yes	131 (75.2%)
Socioeconomic level		No	43 (24.7%)
High	22 (12.6%)	Team sports	
Middle	61 (35.1%)	Yes	94 (54%)
Low	91 (52.3%)	No	37 (21.2%)
Population size		Caregiver	
0–30.000 inhabitants	106 (60.9%)	Yes	18 (10.3%)
>30.000 inhabitants	68 (39%)	No	156 (89.6%)
Type of school		Domestic pet	
Public	155 (89%)	Yes	81 (46.5%)
Concerted or private	19 (10.9%)	No	93 (53.4%)

**Table 3 children-10-00026-t003:** Patient inclusion and exclusion criteria.

Inclusion Criteria	Exclusion Criteria
Age 4–9 years	Children diagnosed with behavioral disorders (autism spectrum disorder, Asperger’s syndrome, Down syndrome, or attention deficit hyperactivity disorder).
Healthy subjects (based on medical history and anamnesis)	
Accompanied by father, mother, or legal guardian	

**Table 4 children-10-00026-t004:** Chi-square test and Cramer’s V applied to the different variables studied and the Wong–Baker faces scale.

Dimensions Studied	*p*-Value	V-Value	Dimensions Studied	*p*-Value	V-Value
Family structure			Habits		
Age	0.001 ***	0.330 ^a^	Bedtime	0.016 *	0.265
Gender	0.305	0.167	Videogames	0.034 *	0.245
Biological offspring	0.646	0.120	Time spent with videogames	0.572	0.169
Order among siblings	0.747	0.127	Time spent watching cartoons	0.797	0.098
Parent marital status	0.311	0.164	Social relations		
Cultural and economic status			After-school activity	0.258	0.174
Socioeconomic level	0.233	0.174	Sports	0.008 **	0.282
Population size	0.345	0.160	Team sports	0.000 ***	0.480 ^a^
Type of school	0.238	0.178	Caregiver	0.690	0.114
			Domestic pet	0.407	0.152

* *p* ≤ 0.05; ** *p* ≤ 0.01; *** *p* ≤ 0.001; ^a^ V > 0.03.

## Data Availability

The study did not report any data.

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
