# Peer review of "Relationship between Children’s Lifestyle and Fear during Dental Visits: A Cross-Sectional Study"

_children, 2022, doi:10.3390/children10010026_

Round 1
Reviewer 1 Report
Comments and suggestions for the submitted article NO children-2082580
1. In the Abstract section (lines 8-9) (and later in the last paragraph of the Introduction section, lines 63-66) the authors have stated that the behaviour of the participants in the dental clinic was observed. This is not true due to the fact that the children were asked to rate their feelings regarding dental treatment by one faces version of analog scales. This statement should be accordingly corrected wherever it appears, and named precisely for what the Wong Baker faces scale is using for.
2. Materials and Methods section has some fundamental problems regarding the fact that it is not completely clear what was evaluated with which kind of scale. For example, the section starts with the mention of the questionnaire consisting of 11 questions, but without emphasizing for what purpose it was used, and without explaining the content of it (lines 68-71). Truly, this questionnaire was linked with the refererence No 22, but this article was originaly published in Spanish language. For the reader it will be very important to know and understand this in order to be able to follow obtained results from it. So, suggestion is to clarify and update this section according to the related subheadings in the next Results section.
Inclusion criteria were not pretty clear because the authors did not state why the children of specific age group (4-9 years) were observed (Table 2, line 110). Was this age interval randomly selected, or just was the case during the period of study research? This should be of the crucial importance due to the pretty various cognitive development of these children, where 4-year olds totaly differ even from 6-year olds, not to mention the oldest 9-year olds in the study sample. So, their answers to the analog scale would totally differ for the same dental stressor, which could cause bias in the collecting of the results. Also, how and why there were two age subgroups subsequently formed, occasionaly, or with some intention? Suggestion is to design the study with children where their normal cognitive and mental capabilities for the observed phenomenon (dental fear and anxiety, DFA) should be similar in the first place, and not to spread them via wide age ranges. So, explain and try to correct the age of participants accordingly.
That is why the main tool for evaluation of the DFA presence (Wong Baker, or any other of this kind) should be pretty big problem. Its normative values (sensitivity, validity) are among the weakest ones compare to other psychometric instruments for evaluation of DFA presence in children. It should be better if the authors could use also some other instrument for evaluation of perhaps children behaviour during dental tretament (which they wrongly stated they did), or perhaps some parental versions of self-rated DFA scales. With this the strength of evaluation of DFA presence among study participants is pretty weak.
Exclusion criteria were also unclear regarding the fact that possible urgent dental treatments were undertaken in the study participants (dental filling/pulp treatment; line 97) If that was the case for any of the child participants, this should also could cause bias in the included patients due to the fact that urgent dental treatments could cause DFA by their innate nature. Please, clarify this, and if there were any, they should be excluded from the total number of participants. Furthermore, it was not stated anywhere if this study research was controlled one or not, how many dentists provided dental treatments to the study participants, were the study research conservative treatment interventions previously planned or not, etc.
Generally, Table 2 consists some of the results that should be moved to this section and not in the methodology, for better following of the related written text and subsequent analyses. There were few place sin this table where the ratios of subgroups (for example offspring, siblings, marital status, SES, etc.) were not similar, but grosly differed, which could probably compromised their statistical analyses.
The authors should state and clarify why regresion analysis was not performed due to the fact that there were many data conducted from the various questionarries, and the opportunity for doing it should not be missed. Only if there were no statistical conditions for performing it, it would be understandable, but then it should be stated as a reason for not doing it. Truly, this matter was briefly mentioned in the study limitations, but not enough strong if the authors could really do this analysis.
From the Results section it was not clear how many children from the specific age groups were in the sample. Truly, there were only exact numbers of study age subgroups mentioned in the Table 2 (61 /35.1%/ vs 113 /64.95/), but there was a big difference between them for a proper statistical analyses. Please, clarify and update these facts.
It is pretty obvious that this submitted paper emphasize the importance of changing the conditions of normal childhood with the various strong influences of modern era, especially those related to the lack of socialization with their and older ages, including adult foreingers. But, the authors should be very careful in drawing the line between the modern difficulties in child socialization and real linkage with the presence and appearance of DFA in children. Also, some of the facts that were collected in questionarries (owing a pet, for example) were pretty unclear to have any scientific kind of relation with DFA presence. The suggestion for authors is therefore to methodologically put new chalenges (as their main intention) into borders of more classic mechanisms of DFA presence and appearance, including the DFA factors as well. If not in the methodology section, they should at least do it in the Discussion section, making theoretical connections between them.
Conclusion section should be updated accordingly to previous comments and suggestions.
Good luck!
Kindest regards.
Author Response
THANK YOU VERY MUCH FOTR YOUR CORRECTIONS, THEY HAVE BEEN VERY IMPORTANT
Point 1: In the Abstract section (lines 8-9) (and later in the last paragraph of the Introduction section, lines 63-66) the authors have stated that the behaviour of the participants in the dental clinic was observed. This is not true due to the fact that the children were asked to rate their feelings regarding dental treatment by one faces version of analog scales. This statement should be accordingly corrected wherever it appears, and named precisely for what the Wong Baker faces scale is using for.
Response 1: Thank you very much fot this correction, we have corrected it, because we have observed the different emotions with Wong Baker faces scale.
Point 2: Materials and Methods section has some fundamental problems regarding the fact that it is not completely clear what was evaluated with which kind of scale. For example, the section starts with the mention of the questionnaire consisting of 11 questions, but without emphasizing for what purpose it was used, and without explaining the content of it (lines 68-71). Truly, this questionnaire was linked with the refererence No 22, but this article was originaly published in Spanish language. For the reader it will be very important to know and understand this in order to be able to follow obtained results from it. So, suggestion is to clarify and update this section according to the related subheadings in the next Results section.
Response 2: We have redacted this section and we think that is more clear now.
Point 3: Inclusion criteria were not pretty clear because the authors did not state why the children of specific age group (4-9 years) were observed (Table 2, line 110). Was this age interval randomly selected, or just was the case during the period of study research? This should be of the crucial importance due to the pretty various cognitive development of these children, where 4-year olds totaly differ even from 6-year olds, not to mention the oldest 9-year olds in the study sample. So, their answers to the analog scale would totally differ for the same dental stressor, which could cause bias in the collecting of the results. Also, how and why there were two age subgroups subsequently formed, occasionaly, or with some intention? Suggestion is to design the study with children where their normal cognitive and mental capabilities for the observed phenomenon (dental fear and anxiety, DFA) should be similar in the first place, and not to spread them via wide age ranges. So, explain and try to correct the age of participants accordingly.
Response 3: We know that fears are evolutionary, they change as the child grows, and his psychobiological system matures, there being a natural tendency for it to progressively disappear. We selected this age interval because children under 4 years of age have more difficulty following instructions, and children older than 9 years are more mature, more independent from the family. and with better attention span. To compare the results between younger and older children (within this selected age range), we grouped them this way. We have redcated and explained it, thank you very much for your correction
Point 4: That is why the main tool for evaluation of the DFA presence (Wong Baker, or any other of this kind) should be pretty big problem. Its normative values (sensitivity, validity) are among the weakest ones compare to other psychometric instruments for evaluation of DFA presence in children. It should be better if the authors could use also some other instrument for evaluation of perhaps children behaviour during dental tretament (which they wrongly stated they did), or perhaps some parental versions of self-rated DFA scales. With this the strength of evaluation of DFA presence among study participants is pretty weak.
Response 4: We've modified that, it was badly augmented, we haven't really assessed the behavior. Thanks.
Point 5: Exclusion criteria were also unclear regarding the fact that possible urgent dental treatments were undertaken in the study participants (dental filling/pulp treatment; line 97) If that was the case for any of the child participants, this should also could cause bias in the included patients due to the fact that urgent dental treatments could cause DFA by their innate nature. Please, clarify this, and if there were any, they should be excluded from the total number of participants. Furthermore, it was not stated anywhere if this study research was controlled one or not, how many dentists provided dental treatments to the study participants, were the study research conservative treatment interventions previously planned or not, etc.
Response 5: We have provided new data to clarify more the procedure.
Point 6: The authors should state and clarify why regresion analysis was not performed due to the fact that there were many data conducted from the various questionarries, and the opportunity for doing it should not be missed. Only if there were no statistical conditions for performing it, it would be understandable, but then it should be stated as a reason for not doing it. Truly, this matter was briefly mentioned in the study limitations, but not enough strong if the authors could really do this analysis.
Response 6: we have already clarified it in the discussion
Point 7: From the Results section it was not clear how many children from the specific age groups were in the sample. Truly, there were only exact numbers of study age subgroups mentioned in the Table 2 (61 /35.1%/ vs 113 /64.95/), but there was a big difference between them for a proper statistical analyses. Please, clarify and update these facts.
Response 7: Thank you very much, we have clarified it in the discussion.
Point 8: It is pretty obvious that this submitted paper emphasize the importance of changing the conditions of normal childhood with the various strong influences of modern era, especially those related to the lack of socialization with their and older ages, including adult foreingers. But, the authors should be very careful in drawing the line between the modern difficulties in child socialization and real linkage with the presence and appearance of DFA in children. Also, some of the facts that were collected in questionarries (owing a pet, for example) were pretty unclear to have any scientific kind of relation with DFA presence. The suggestion for authors is therefore to methodologically put new chalenges (as their main intention) into borders of more classic mechanisms of DFA presence and appearance, including the DFA factors as well. If not in the methodology section, they should at least do it in the Discussion section, making theoretical connections between them.
Response 8: Thank you very much, we have connected classic mechanism with new factors in the discussion. W think that is better now.
Point 9: Conclusion section should be updated accordingly to previous comments and suggestions.
Response 9: Thank you very much, it is accordingly with them.
Kindest regards.
Reviewer 2 Report
The manuscript is very well written and after a few minor revisions could be suitable for publication.
Firstly, the Wong Baker faces scale was shown to the children after the conservative treatment, in which the types of restorations were not disclosed by the authors. The treatment may range from one-surface anterior composite restoration to the most complex dental pulp therapy. The scores may also be influenced by the severity of dental pain experienced by the children during the visit, the number of treatments, as well as the personal attributes of the dental provider(s)/staff, and other confounding factors. These factors may introduce bias in the results.
Secondly, in the results section, referring to Table 2 there's a * symbol on socioeconomic status, but no notes are given.
Finally, Section 3.2 stated that; “Patients who belong to families of medium (34.7%) or low socioeconomic level (65.2%) produced many more sad or incredibly sad responses than in the group of high socioeconomic level (0%)”. However, when referring to the percentages of the socioeconomic level in Table 2, the percentages are different i.e : High=22 (12.6%); Middle =61 (35.1%) ;Low =91 (52.3%).
Thank you.
Author Response
Thank you very much for your appreciations
Point 1: Firstly, the Wong Baker faces scale was shown to the children after the conservative treatment, in which the types of restorations were not disclosed by the authors. The treatment may range from one-surface anterior composite restoration to the most complex dental pulp therapy. The scores may also be influenced by the severity of dental pain experienced by the children during the visit, the number of treatments, as well as the personal attributes of the dental provider(s)/staff, and other confounding factors. These factors may introduce bias in the results.
Response 1: Thank you very much, it is right, we have included it as limitation of the study
Point 2: Secondly, in the results section, referring to Table 2 there's a * symbol on socioeconomic status, but no notes are given.
Response 2: Sorry, we have already eliminated it.
Point 3: Finally, Section 3.2 stated that; “Patients who belong to families of medium (34.7%) or low socioeconomic level (65.2%) produced many more sad or incredibly sad responses than in the group of high socioeconomic level (0%)”. However, when referring to the percentages of the socioeconomic level in Table 2, the percentages are different i.e : High=22 (12.6%); Middle =61 (35.1%) ;Low =91 (52.3%).
Response 3: Yes, the percentages of table 2 are the frequencies of the variables, thus they don´t coincide with the results.
Round 2
Reviewer 1 Report
Dear authors,
good luck with the submission.
Author Response
Thanks